# Comprehensive Evaluation of Multiple Approaches Targeting ABCB1 to Resensitize Docetaxel-Resistant Prostate Cancer Cell Lines

**DOI:** 10.3390/ijms24010666

**Published:** 2022-12-30

**Authors:** Dinah Linke, Lukas Donix, Claudia Peitzsch, Holger H. H. Erb, Anna Dubrovska, Manuel Pfeifer, Christian Thomas, Susanne Fuessel, Kati Erdmann

**Affiliations:** 1Department of Urology, Faculty of Medicine, University Hospital Carl Gustav Carus, Technische Universität Dresden, 01307 Dresden, Germany; 2National Center for Tumor Diseases (NCT), Partner Site Dresden, 01307 Dresden, Germany; 3German Cancer Research Center (DKFZ), 69120 Heidelberg, Germany; 4OncoRay—National Center for Radiation Research in Oncology, Faculty of Medicine, University Hospital Carl Gustav Carus, Technische Universität Dresden, Helmholtz-Zentrum Dresden-Rossendorf (HZDR), 01309 Dresden, Germany; 5Center for Regenerative Therapies Dresden (CRTD), 01307 Dresden, Germany; 6German Cancer Consortium (DKTK), Partner Site Dresden, 01307 Dresden, Germany; 7Institute of Radiooncology—OncoRay, Helmholtz-Zentrum Dresden-Rossendorf (HZDR), 01309 Dresden, Germany; 8Institute of Legal Medicine, Faculty of Medicine, University Hospital Carl Gustav Carus, Technische Universität Dresden, 01307 Dresden, Germany

**Keywords:** ABCB1, chemoresistance, docetaxel, elacridar, glycosylation, P-glycoprotein, prostate cancer, siRNA, tariquidar, tunicamycin

## Abstract

Docetaxel (DTX) is a mainstay in the treatment of metastatic prostate cancer. Failure of DTX therapy is often associated with multidrug resistance caused by overexpression of efflux membrane transporters of the ABC family such as the glycoprotein ABCB1. This study investigated multiple approaches targeting ABCB1 to resensitize DTX-resistant (DTXR) prostate cancer cell lines. In DU145 DTXR and PC-3 DTXR cells as well as age-matched parental controls, the expression of selected ABC transporters was analyzed by quantitative PCR, Western blot, flow cytometry and immunofluorescence. ABCB1 effluxing activity was studied using the fluorescent ABCB1 substrate rhodamine 123. The influence of ABCB1 inhibitors (elacridar, tariquidar), ABCB1-specific siRNA and inhibition of post-translational glycosylation on DTX tolerance was assessed by cell viability and colony formation assays. In DTXR cells, only ABCB1 was highly upregulated, which was accompanied by a strong effluxing activity and additional post-translational glycosylation of ABCB1. Pharmacological inhibition and siRNA-mediated knockdown of ABCB1 completely resensitized DTXR cells to DTX. Inhibition of glycosylation with tunicamycin affected DTX resistance partially in DU145 DTXR cells, which was accompanied by a slight intracellular accumulation and decreased effluxing activity of ABCB1. In conclusion, DTX resistance can be reversed by various strategies with small molecule inhibitors representing the most promising and feasible approach.

## 1. Introduction

The semi-synthetic taxane docetaxel (DTX) is widely used as a chemotherapeutic agent for the treatment of several of the most prevalent carcinomas including head and neck, breast, non-small cell lung, gastric and prostate cancer (PCa). In PCa, DTX represents a treatment option for metastatic hormone-sensitive and for castration-resistant prostate cancer (mCRPC) [1]. Furthermore, the taxane cabazitaxel (CTX) can be used as second-line therapy for mCRPC [1]. Although DTX improved the overall survival in mCRPC patients [2,3], the duration of successful treatment is limited by the emergence of resistance. To study chemoresistance in PCa cells, DTX-resistant (DTXR) cells of the mCRPC-derived cell lines DU145 and PC-3, which are negative for the androgen receptor (AR), were generated by dose escalation with DTX [4]. In these DTXR cells, DTX tolerance is highly increased compared to their age-matched parental control cells (CTRL). Previous results from our group demonstrated that the half-maximal inhibitory concentration (IC_50_) values for DTX were increased about 70-fold in DU145 DTXR cells and about 40-fold in PC-3 DTXR cells compared to their parental CTRL cells, respectively [5].

Taxanes mediate cytotoxicity by stabilizing microtubules resulting in the disassembly of the mitotic spindle during the M phase, which in turn causes mitotic arrest followed by apoptosis [6]. The cellular adaptations to taxane exposure vary between cancer entities and include decreased expression of taxane-importing membrane transporters, preferred expression of tubulin isotypes less susceptible to taxane binding, apoptotic escape and increased expression of taxane-effluxing membrane transporters [6]. Drug-effluxing membrane transporters of the ATP-binding cassette (ABC) transporter family consist of seven subfamilies encoded by 49 genes [7]. Under physiologic conditions, ABC transporters are expressed in healthy tissue with secretory or barrier function such as the intestinal epithelium, renal glomeruli or the blood-brain barrier, where they help to exclude harmful small molecules at the tissue as well as at the single cell level [8]. Some ABC transporters feature broad substrate specificities and can efflux numerous structurally and biochemically unrelated compounds [9], thereby playing a central role in mediating multidrug resistance (MDR) by limiting intracellular drug concentrations [6]. ABCB1 and ABCG2 (BCRP) are two of the most extensively studied members of the ABC transporter family involved in MDR [8,10,11]. ABCB1—also known as multidrug resistance protein 1 (MDR1) and P-glycoprotein (P-gp)—has been linked to taxane resistance in various cancer entities including mCRPC [11,12,13,14]. Other members of the ABC transporter family involved in taxane resistance are ABCA3, ABCB4, ABCB11, ABCC1, ABCC2 and ABCC10 [10,11]. In addition, ABCG2 is an important stem cell marker in PCa [15].

Consequently, targeting ABC transporters, such as ABCB1, to overcome drug resistance in cancer patients has been explored in clinical trials in various cancer entities [16,17,18,19,20]. These studies tested combinations of cytotoxic treatment and the unspecific ABC transporter inhibitor verapamil [21]. Adverse events, including hematologic toxicity, were dose-limiting. Subsequently, compounds for more specific targeting of ABCB1—such as elacridar and tariquidar—were developed. In vitro, mainly elacridar has been investigated in various taxane-resistant PCa cells, in which a cell line-specific resensitization against DTX or CTX could be observed when ABCB1 was pharmacologically inhibited [13,22,23,24]. However, clinical trials investigating elacridar and tariquidar in various cancers, but not PCa so far, yielded mixed results [25,26,27,28,29,30,31].

As mentioned before, the ABCB1 protein is also called P-glycoprotein due to the post-translational addition of N-linked glycans to three glycosylation motifs in the first extracellular loop [32,33]. N-linked glycosylation is a multi-step process. First, a core glycosylation consisting of a glucose(3)-mannose-(9)-N-acetylglucosamine(2) (Glc_3_Man_9_GlcNAc_2_) residue is added in the endoplasmic reticulum (ER) [34,35]. Then, additional saccharides of varying complexity are added in the Golgi apparatus’s core as the maturing glycoprotein is headed towards its final destination [34]. In the case of ABCB1, glycosylation accounts for a significant proportion of the protein’s molecular weight. Non-glycosylated ABCB1 has a molecular weight of 130 kDa, the core-glycosylated protein of 150 kDa and the fully glycosylated protein of 170 kDa [36]. However, the details of how exactly the glycosylation contributes to the functionality of ABCB1 are not entirely elucidated to date.

This study evaluated the expression of eight ABC transporters associated with taxane resistance and/or MDR in mCRPC-derived DU145 and PC-3 DTXR and CTRL cells. Ultimately, ABCB1 was identified as the dominant ABC transporter highly upregulated in DU145 and PC-3 DTXR cells, whereas a significant involvement of other ABC transporters was unlikely. Therefore, different approaches targeting ABCB1 to overcome DTX resistance, including treatment with ABCB1 inhibitors and siRNA-mediated knockdown of *ABCB1*, were evaluated. Since the ABCB1 protein exhibited additional glycosylation in the DTXR cells, the influence of the protein’s glycosylation status on its cellular localization and effluxing activity as well as on the DTX resistance was also investigated. Overall, this study constitutes a comprehensive evaluation of multiple approaches targeting ABCB1 to resensitize DTXR PCa cell lines.

## 2. Results

### 2.1. DTX-Resistant DU145 and PC-3 Cells Express High Levels of ABCB1

We previously demonstrated that the IC_50_ values for DTX increased from 5.9 to 388 nM in DU145 DTXR and from 8.2 to 305 nM in PC-3 DTXR cells compared to their parental CTRL cells [5]. Therefore, the expression of eight ABC transporters associated with taxane resistance and/or MDR [8,10,11] was evaluated by quantitative PCR (qPCR) in DU145 and PC-3 DTXR and CTRL cells (Figure 1a). Of the selected ABC transporters, ABCG2 is also a known cancer stem cell marker in PCa and was analyzed in more detail [15]. Most of the investigated ABC transporter genes (*ABCB4*, *B11*, *C2*, *C10*, *G2*) were expressed at very low levels in both CTRL and DTXR cells with crossing points (CP) close to or even beyond the detection threshold (CP ≥ 35; Figure 1a). Although partly expressed at rather low levels in both cell types, transporters of the C-family (*ABCC1*, *C2* and *C10*) were downregulated in DTXR cells compared to the respective CTRL cells. In contrast, *ABCA3* was moderately expressed at similar levels in CTRL and DTXR cells. *ABCB1*, however, exhibited very low expression levels (CP ≥ 35) in the CTRL cells, but was highly upregulated in DTXR cells (Figure 1a). Western blot analysis confirmed the highly increased expression of ABCB1 in the DTXR cells at the protein level compared to the respective CTRL cells (Figure 1b). Interestingly, both DU145 and PC-3 DTXR cells showed protein bands with a higher molecular weight and a broader range between ~150–170 kDa, whereas only a slight protein band with a molecular weight of ~150 kDa was observed in CTRL cells. The difference in molecular weight might be caused by distinct glycosylation of the ABCB1 protein in CTRL and DTXR cells. In contrast to ABCB1, low levels of ABCG2 protein were detected in all cell lines without differential expression between CTRL and DTXR cells (Figure 1b).

Additionally, ABCB1 and ABCG2 protein levels were investigated by flow cytometry, which confirmed the strong upregulation of ABCB1 in DTXR cells compared to CTRL cells (Figure 1c). In DU145 and PC-3 DTXR cells, the fluorescence intensities for ABCB1 protein were significantly elevated by 4.2- and 5.3-fold compared to the respective CTRL cells (Figure 1c). Accordingly, DTXR cells exhibited a higher percentage of ABCB1-positive cells compared to CTRL cells (DU145: 88.1 ± 7.8% vs. 9.9 ± 7.3%; PC-3: 98.5 ± 1.2% vs. 26.7 ± 20.6%; *p* < 0.01, non-paired t-test with Welch’s correction). Although most of the CTRL and DTXR cells were slightly positive for ABCG2, both cell types showed similar fluorescence intensities for ABCG2 and thus, no differential expression could be detected (Figure 1d).

Taken together, our data show that ABCB1 is the dominant ABC transporter expressed in DU145 and PC-3 DTXR cells.

### 2.2. ABCB1 Inhibitors Reverse DTX Resistance in DU145 and PC-3 Cells

In order to investigate the ABCB1 efflux activity, DU145 and PC-3 cells were incubated with the fluorescent substrate rhodamine 123 (Rh123) [37] in the presence or absence of the inhibitors elacridar and tariquidar. According to preliminary dose-response experiments, these inhibitors were used in the non-toxic, but effective concentration of 50 nM (Appendix A). In DU145 and PC-3 CTRL cells, intracellular Rh123 fluorescence intensity decreased by at most 40% within 60 min after removal of treatment, regardless of whether ABC transporter inhibitors were present or not (Figure 2a,b). Without inhibitors, no intracellular Rh123 signal was observed in DU145 and PC-3 DTXR cells, suggesting a highly effective effluxing activity of ABCB1. In the presence of either elacridar or tariquidar, Rh123 efflux in DU145 and PC-3 DTXR cells was diminished markedly but with varying efficacy (Figure 2a,b). After 60 min, elacridar- and tariquidar-treated PC-3 DTXR cells showed nearly the same fluorescence intensity as the CTRL cells with the corresponding treatment (Figure 2b): 92% (DTXR) vs. 89% (CTRL) for elacridar-treated and 54% (DTXR) vs. 64% (CTRL) for tariquidar-treated cells, respectively. In contrast, marked differences in Rh123 fluorescence after 60 min could be observed between DU145 CTRL and DTXR cells (Figure 2a): 27% (DTXR) vs. 70% (CTRL) for elacridar-treated and 12% (DTXR) vs. 61% (CTRL) for tariquidar-treated cells, respectively. However, the intracellular Rh123 fluorescence at 60 min was either significantly or per trend higher in DU145 DTXR and PC-3 DTXR cells treated with elacridar or tariquidar compared to DTXR cells without ABCB1 inhibitor (Figure 2a,b). Furthermore, the accumulation of Rh123 was more constant over time in PC-3 cells (Figure 2b) than in DU145 cells (Figure 2a). Of note, ABCB1 inhibition with elacridar resulted in a more durable inhibition of Rh123 efflux compared to tariquidar in both cell lines.

Next, we investigated whether ABCB1 inhibition can resensitize DTXR cells and for this purpose, cells were treated with serial dilutions of DTX alone or in combination with either elacridar or tariquidar (Figure 3a). Both inhibitors achieved a complete resensitization of DU145 DTXR and PC-3 DTXR cells towards DTX. Based on both the WST-1 (Figure 3a) and the crystal violet assay (Appendix A), IC_50_ values of the CTRL cells remained the same with or without either of the inhibitors, whereas the IC_50_ values for DTX in DTXR cells decreased significantly to the IC_50_ levels of the respective CTRL cells (Figure 3a and Appendix A).

To confirm the resensitization of DTXR cells through ABCB1 inhibitors with another biological read-out, cell colony formation assays were conducted (Figure 3b). The ability of a single cell to form colonies (clonogenic survival) is a unique feature of cancer cells and provides information about long-term proliferation [38]. Since elacridar showed a more potent ABCB1 inhibition in the Rh123 efflux experiment, this drug was used exemplarily for the colony formation assay. Therefore, cells were treated with either elacridar alone, DTX alone or a combination of both. Elacridar alone did not affect the relative number of colonies compared to untreated cells. While DTX alone (10 nM) significantly inhibited colony formation in CTRL cells, it did not affect colony formation in DTXR cells. In CTRL cells, the treatment with elacridar together with DTX had no further effect on the clonogenic survival. In contrast, in DTXR cells, the combination of elacridar and DTX diminished the relative number of colonies significantly to almost zero (Figure 3b).

### 2.3. Specific Knockdown of ABCB1 Is Sufficient to Resensitize DTX-Resistant DU145 and PC-3 Cells

As another approach to overcome DTX resistance, a specific siRNA (siR-ABCB1) was used to transiently knockdown *ABCB1*. Due to the increased level of ABCB1 mRNA and protein in DTXR cells, the *ABCB1* knockdown was performed only in DTXR, but not in CTRL cells. At first, successful downregulation of *ABCB1* was verified by qPCR and Western blot analysis. Following knockdown, the mRNA expression and protein amount of ABCB1 were significantly diminished by up to 85% compared to cells treated with a control construct (siR-CON; Figure 4a).

After verification of the *ABCB1* knockdown, its influence on the DTX resistance was investigated. For this, DTXR cells were treated with a serial dilution of DTX 96 h after the start of transfection. The IC_50_ values of DTX based on a WST-1 assay were decreased in siR-ABCB1-treated compared to siR-CON-treated DTXR cells with 4.0 nM vs. 701.5 nM for DU145 DTXR and 5.2 vs. 75.8 nM for PC-3 DTXR cells, respectively (Figure 4b). However, the observed reduction of the IC_50_ values reached no statistical significance compared to cells treated with DTX and siR-CON. Similar results were observed with the crystal violet assay (Appendix A).

In addition, colony formation assays were performed to examine the effect of the *ABCB1* knockdown on the DTX resistance (Figure 4c). Treatment with siR-ABCB1 alone had no negative effect on the clonogenic potential compared to the siR-CON-treated cells, indicating that cells with downregulated ABCB1 protein are not limited in their ability to form new colonies (Figure 4c). DTX in combination with siR-CON reduced the number of colonies by 25% in DU145 DTXR cells but had nearly no effect on colony growth in PC-3 DTXR cells compared to siR-CON treatment alone. In contrast, the combination of siR-ABCB1 and DTX significantly diminished clonogenic survival by 58% in DU145 DTXR and 91% in PC-3 DTXR cells compared to siR-CON-treated cells (Figure 4c).

### 2.4. Tunicamycin Treatment Leads to Deglycosylation of the ABCB1 Protein in DU145 and PC-3 DTXR Cells

To prove that the protein bands of higher molecular weight in DTXR cells compared to CTRL cells (Figure 1b) were caused by glycosylation of the ABCB1 protein, cell lysates were incubated with the peptide-N-glycosidase F (PNGase F) for 2 h, which cleaves off N-glycans from the mature protein. This deglycosylation resulted in a decreasing signal of the upper protein band (~170 kDa) and the appearance of a new protein band (~130 kDa) representing the unglycosylated ABCB1 protein (Figure 5a). The Western blot was then analyzed densitometrically in order to determine the amount of fully glycosylated, core-glycosylated and unglycosylated protein as part of the total ABCB1 protein amount in % (Appendix A and Appendix A). The analysis yielded an increase of unglycosylated protein from about 3 to 75% of the total protein amount in DU145 DTXR and from 5 to 60% in PC-3 DTXR cells, whereas only about 10% of fully glycosylated ABCB1 protein remained in both cell lines (Figure 5a). Of note, untreated DU145 DTXR cells exhibited a higher amount of fully glycosylated ABCB1 protein (~85%) than PC-3 DTXR cells (~70%; Figure 5a).

After confirming the strong glycosylation of the ABCB1 protein in DTXR cells, the next step was to investigate whether glycosylation plays a role in maintaining the DTX resistance. For this purpose, the small molecule inhibitor tunicamycin was used, which inhibits the glycosylation at the first step leading to an unglycosylated protein. Based on dose-response experiments and Western blot analysis (Appendix A), 100 ng/mL tunicamycin was chosen as an optimal treatment concentration. To achieve deglycosylation, cells were pre-treated with 100 ng/mL tunicamycin for 96 h. Consequently, a shift to protein bands with lower molecular weights was observed in the Western blot indicating the deglycosylation of ABCB1 protein (Figure 5b). However, tunicamycin treatment of live cells was less effective than the deglycosylation with PNGase F, which was performed with harvested protein lysates (Figure 5a). According to the densitometrical analysis, more than 50% of ABCB1 protein was still fully glycosylated in both DU145 DTXR and PC-3 DTXR cells pre-treated with tunicamycin (Figure 5b). In DU145 DTXR cells, the amount of unglycosylated protein only increased from about 4 to 22% of the total protein amount after tunicamycin treatment (Figure 5b). In contrast, the deglycosylation was even less effective in PC-3 DTXR cells with a marginal increase of unglycosylated ABCB1 protein from 2 to 8% (Figure 5b). Since discontinued glycosylation of proteins may lead to incorrect folding and subsequent proteasomal degradation [39], the total amount of ABCB1 protein was also determined. However, the total ABCB1 amount in tunicamycin-treated DU145 DTXR and PC-3 DTXR cells did not significantly differ from untreated cells (Appendix A). This suggests that deglycosylation by tunicamycin was not substantially associated with proteasomal degradation of the ABCB1 protein. In order to confirm this finding, DU145 and PC-3 DTXR cells were treated with a combination of tunicamycin and the proteasome inhibitor MG132 [40], since proteasomes are involved in the degradation of unneeded or damaged proteins via proteolysis. Compared to treatment with tunicamycin alone, the combination of tunicamycin and MG132 did not result in an increased amount of ABCB1 protein in DU145 and PC-3 DTXR cells, confirming the absence of proteasomal degradation (Appendix A).

To evaluate the influence of the ABCB1 protein’s glycosylation status on its cellular localization, immunofluorescence staining of ABCB1 was performed with DU145 and PC-3 CTRL and DTXR cells as well as with DTXR cells pre-treated with 100 ng/mL tunicamycin for 96 h. Compared to the CTRL cells, the fluorescence signal in DU145 and PC-3 DTXR cells was very strong (Figure 5c), which indicates a high protein level of ABCB1 and is consistent with the previous expression data (Figure 1a,b). Furthermore, the ABCB1 protein was mainly detected in the cell membrane of the DTXR cells (Figure 5c). Pre-treatment of DTXR cells with 100 ng/mL tunicamycin for 96 h resulted in a more diffuse fluorescence signal distributed in the whole cell as compared to the cell membrane-concentrated signal seen in DTXR cells without tunicamycin pre-treatment. This effect was more prominent in DU145 DTXR than in PC-3 DTXR cells.

### 2.5. Deglycosylation by Tunicamycin Reduces DTX Tolerance in DU145 DTXR Cells but Not in PC-3 DTXR Cells

In order to investigate the influence of the ABCB1 glycosylation status on the ABCB1 efflux activity, DU145 and PC-3 DTXR cells with or without tunicamycin pre-treatment for 96 h were incubated with Rh123 followed by a longitudinal assessment of intracellular fluorescence over 60 min. DU145 DTXR cells pre-treated with tunicamycin exhibited a markedly diminished Rh123 efflux with significantly higher intracellular amounts of Rh123 at 0, 10 and 20 min compared to DU145 DTXR cells without pre-treatment (Figure 6a). In contrast, PC-3 DTXR cells pre-treated with tunicamycin only showed a higher intracellular Rh123 fluorescence at the start of the measurements (t_0_) compared to PC-3 DTXR cells without pre-treatment (Figure 6b).

Next, DU145 and PC-3 DTXR cells with or without pre-culturing with tunicamycin for 96 h were treated with a serial dilution of DTX in order to study the effect of deglycosylation on the DTX tolerance. The dose-response curves based on a WST-1 assay showed a reduction of the DTX tolerance in pre-treated DU145 DTXR cells compared to DU145 DTXR cells without tunicamycin pre-treatment (Figure 7a). Furthermore, the calculated IC_50_ values for DTX were markedly reduced by about 50% after pre-treatment with tunicamycin (from 192.7 nM to 96.1 nM). In contrast, no differences between PC-3 DTXR cells with and without tunicamycin pre-treatment could be observed regarding their tolerance towards DTX (Figure 7a). The differential influence of tunicamycin pre-treatment on the DTX tolerance in DU145 and PC-3 DTXR cells could also be confirmed by the crystal violet assay with significant differences for DU145 DTXR cells (Figure 7b).

## 3. Discussion

Resistance against taxanes, such as DTX, CTX and paclitaxel (PTX), is mainly associated with higher cellular efflux mediated by various ABC transporters [10,11]. Particularly, ABCB1 and ABCG2 are known to mediate MDR against a variety of structurally different anticancer drugs [8,10,11] with ABCG2 also being a well-recognized stem cell marker in PCa [15]. In this study, eight selected ABC transporters known to be involved in taxane resistance and/or MDR were investigated for their expression in mCRPC-derived DU145 and PC-3 DTXR and CTRL cells. Among the investigated ABC transporters, only ABCB1 was highly upregulated at the transcript and protein level in DU145 and PC-3 DTXR cells compared to the respective CTRL cells. These results align with those previously published by Seo et al. [14], who also generated DTXR sublines of DU145, PC-3 and CWR22 PCa cells by dose escalation. They performed RNA sequencing and observed that ABCB1 mRNA was upregulated 9.7-fold to 12-fold compared to the DTX-naïve CTRL cells. Except for the two-fold upregulation of ABCB4 and ABCB8 as well as the downregulation of ABCB11, ABCC2 and ABCC12, most other ABC transporters exhibited no differential expression in the DTXR cells [14]. In line with the data published by Seo et al., ABCG2—another important MDR protein and PCa stem cell marker– was not differentially expressed in the DTXR sublines of the present study. Takeda et al. also showed a distinct upregulation of ABCB1 in PTX-resistant DU145 and PC-3 cells, which was partly linked to the demethylation of the *ABCB1* promoter [12]. Furthermore, the generation of dual DTX-CTX-resistant DU145 and C4-2B sublines was accompanied by augmented levels of ABCB1 compared to DTXR cells [24].

The expression differences between CTRL and DTXR cells suggest that the resistant cells need to employ several adjustments to gene expression and protein maturation to establish and sustain elevated levels of ABCB1 at the cell membrane. In accordance with the differential ABCB1 mRNA expression, highly elevated protein levels were confirmed in DTXR cells with strong protein bands at ~170 and 150 kDa, whereas only a weak protein band at ~150 kDa was detected in CTRL cells. Based on the experiments inhibiting glycosylation, the increased molecular weight in DTXR cells could be attributed to the protein’s glycosylation. In contrast, Seo et al. and Takeda et al. did not detect ABCB1 protein in their CTRL cell lines and only observed a single protein band at around 170 kDa in their taxane-resistant cells [12,14]. In the present study, however, siRNA-mediated knockdown confirmed the detected protein bands as ABCB1 protein. Compared to the CTRL cells, the ABCB1 protein was mainly located in the cell membrane in both DTXR sublines, which is in line with the findings by Lima et al. for DTXR LNCaP and C4-2B PCa cells [41,42]. Furthermore, the overexpression of ABCB1 in DTXR cells was accompanied by a high effluxing rate of the ABCB1 substrate Rh123, which was also demonstrated in C4-2B DTXR PCa cells [41,42].

In order to address ABCB1-mediated DTX resistance in DU145 and PC-3 DTXR cells, multiple approaches targeting ABCB1 were evaluated in this study. A complete resensitization of the DTXR cells against DTX to the level of CTRL cells was achieved using the ABCB1 inhibitors elacridar and tariquidar, which further emphasizes the improbability that other ABC transporters are involved in mediating DTX resistance in the investigated cell line models. The restored DTX tolerance was accompanied by a profound inhibition of clonogenic survival and efflux activity. Of note, efflux activity inhibition with elacridar was more effective and durable than with tariquidar, validating the higher potency of elacridar [43]. Overall, our findings align with those of several other groups that were able to resensitize various taxane-resistant PCa cell lines overexpressing ABCB1 by using ABCB1 inhibitors [13,22,23,24,44]. The majority of the published studies used elacridar in a concentration of 250–500 nM to inhibit ABCB1 [13,22,23,24], which is five to ten times higher than the concentration of 50 nM used in the present work.

In the study by O’Neill et al., treatment with 250 nM elacridar could reverse DTX resistance in DU145 and 22RV1 DTXR cells with upregulated ABCB1. In contrast, their PC-3 DTXR cells did not express ABCB1 and consequently, treatment with 500 nM elacridar failed to resensitize these cells. The authors identified other mechanisms as possible causes for DTX resistance in these PC-3 DTXR cells, such as the activation of the transcription factor NF-κB and changes in the apoptotic phenotype [23]. In addition, the reversal of taxane resistance by elacridar was shown for DTX- and dual DTX-CTX-resistant DU145 and C4-2B PCa cells [13,22,24]. To date, only the study by Shimizu et al. investigated the effect of tariquidar on PCa cell lines. They were able to resensitize LNCaP95 DTXR cells by treatment with 50 nM tariquidar [44], which equals the concentration used in the present study.

Despite dynamic research efforts to bring ABCB1 inhibitors into clinical use, none has been approved for therapeutic application to date [21,25,26,27,28,29,30,31]. In several clinical studies (phase I and II), tariquidar seemed to be a promising candidate for overcoming DTX resistance. A phase III study (NCT00042302) was already initiated testing a combination of tariquidar with PTX and carboplatin or vinorelbine as first-line therapy in non-small-cell lung cancer. Unfortunately, this trial needed to be aborted due to increased toxicity [21]. For elacridar, some encouraging phase I studies were conducted. However, further clinical development was stopped due to increased systemic effects of DTX after another phase I combination study with DTX and epirubicin [21]. These results are not surprising as ABCB1 plays a paramount physiological role in the efflux of detrimental compounds across membranes to protect the most sensitive and critical tissues in the body from toxins [8]. Therefore, innovative new ways for targeting ABCB1 and other ABC transporters are needed. In addition to repurposing established drugs, a promising approach may be examining a new generation of ABC transporter inhibitors, including natural products, lipids and surfactants, peptidomimetics or dual ligands as well as nanocarriers for targeted drug delivery [21,45].

Furthermore, this study investigated the siRNA-mediated knockdown of *ABCB1* as another approach to overcome DTX resistance. Treatment with a specific siRNA against *ABCB1* diminished the IC_50_ values of DTX in both DTXR sublines to the level of CTRL cells. However, a higher concentration of DTX was needed to effectively inhibit clonogenic survival of siRNA-transfected DTXR cells compared to co-treatment with elacridar (50 vs. 10 nM). Similar to our results, siRNA-mediated knockdown of *ABCB1* in PTX-resistant DU145 cells restored PTX sensitivity with decreasing IC_50_ values for PTX by about eight-fold from 537.9 nM (non-targeting siRNA) to 60.8 nM (ABCB1-targeting siRNA) [12]. However, in contrast to our results, this study did not observe any significant effect of the *ABCB1* knockdown on PTX tolerance in PTX-resistant PC-3 cells (IC_50_ 198.4 vs. 140.6 nM) although *ABCB1* was upregulated [12]. This, in turn, indicates that the upregulation of *ABCB1* might not always be the main cause of taxane resistance.

Albeit the glycosylation of ABCB1 has been investigated in several studies [32,39,46,47], to the best of our knowledge, there are none evaluating the relevance of ABCB1 glycosylation in PCa cell lines so far. In the literature, the influence of glycosylation on the function of ABCB1 is considered contradictory. Kramer et al. showed that deglycosylation with tunicamycin led to an increased accumulation of the cytostatic drug daunorubicin in a human colon carcinoma cell line, which was explained by the loss in ABCB1 function [48]. Wojtowicz et al. made similar observations revealing reduced cell viability in drug-resistant ovarian and colorectal cancer cells after co-treatment with tunicamycin and various chemotherapeutic drugs including PTX [49]. Using immunofluorescence staining after tunicamycin treatment, they also observed less ABCB1 protein localized at the cell membrane as it accumulated in the cytoplasm [49], which is in line with our findings for DU145 DTXR cells.

Other studies suggest that the glycosylation is not per se required for the functioning and activity of ABCB1 [32,47]. Albeit the transfection of drug-sensitive human melanoma cells with wild-type ABCB1 was significantly more efficient than the transfection with glycosylation-deficient variants, both approaches yielded drug-resistant cells [32]. The authors hypothesized that the glycosylation is required for the correct trafficking of the protein to the outer cell membrane and/or its membrane stabilization [32]. In accordance, a glycosylation-deficient variant of ABCB1 showed a reduced membrane localization compared to the wild-type protein after expression in HeLa cells [47]. However, an additional study showed that the inhibition of the glycosylation with tunicamycin did not reduce the cell membrane localization or the transport activity of ABCB1 in L1210 cells [50].

Based on this literature review, it can be assumed that whether MDR can be reversed by deglycosylation of ABCB1 depends on the choice of the (tumor) cell model. We provide the first report on this issue in DU145 and PC-3 DTXR cells. First, we could show that ABCB1 is highly glycosylated in these cells and that treatment with the glycosylation inhibitor tunicamycin led to an increased amount of deglycosylated protein, which was more pronounced in DU145 DTXR than in PC-3 DTXR cells. In tunicamycin-treated DU145 DTXR cells, an intracellular accumulation but no proteasomal degradation of ABCB1 protein was observed, which was accompanied by a significantly diminished ABCB1 efflux activity. Furthermore, the DTX tolerance in DU145 DTXR cells pre-treated with tunicamycin for 96 h was markedly reduced compared to untreated DU145 DTXR cells. It can be assumed that the reduced amount of ABCB1 protein in the cell membrane is responsible for the decreased drug efflux. However, the IC_50_ values of DTX did not reach the same level as after treatment with the ABCB1 inhibitors or after *ABCB1* knockdown. This might be explained by the incomplete deglycosylation following tunicamycin treatment with more than 50% of ABCB1 protein still being fully glycosylated. Additionally, it has to be noted that tunicamycin does not specifically deglycosylate only the ABCB1 protein but all glycoproteins in the cell. Thus, other cellular mechanisms might be affected by the treatment with tunicamycin. In order to confirm and further investigate the influence of the ABCB1 glycosylation status on its efflux activity, experiments with glycosylation-deficient ABCB1 variants should be conducted in the future.

In contrast to DU145 DTXR cells, tunicamycin-mediated deglycosylation of ABCB1 was less effective in PC-3 DTXR cells and did not lead to significant differences in ABCB1 efflux activity, DTX tolerance or cellular localization of the ABCB1 protein. This suggests again that additional mechanisms might be involved in DTX resistance of PC-3 cells. As mentioned before, O’Neill et al. also attributed the DTX resistance of their PC-3 cells to other mechanisms than upregulation of ABCB1 [23]. Furthermore, the different p53 status of DU145 (p53-mutant) and PC-3 (p53-negative) cells [51] could explain the differential effect of tunicamycin on DU145 DTXR and PC-3 DTXR cells. Previously, it has been shown that mutant p53 enhances the efficiency of the N-glycoprotein folding machinery in various cancer cells [52]. Therefore, it could be possible that the ABCB1 protein is N-glycosylated more effectively in p53-mutant DU145 DTXR compared to p53-negative PC-3 DTXR cells. Conversely, tunicamycin-mediated deglycosylation of the ABCB1 protein might be more effective in DU145 DTXR than in PC-3 DTXR cells. Accordingly, we could show that DU145 DTXR cells (i) exhibited a higher amount of fully glycosylated ABCB1 protein and (ii) deglycosylation by tunicamycin was more effective compared to PC-3 DTXR cells.

As mentioned above, there is still a lot of glycosylated and core-glycosylated ABCB1 protein left in the DU145 and PC-3 DTXR cells, although both cell lines were pre-cultured with tunicamycin for 96 h with the non-toxic dose of 100 ng/mL. These findings suggest that glycosylated ABCB1 protein is very stable in the cell membrane in these cell lines. The half-life of the ABCB1 protein is estimated to be 48–72 h [48]. Apart from possible roles in ABCB1 folding and trafficking, glycosylation may also play an essential role in stabilizing ABCB1 at the cell membrane by interacting with galectins, which preferentially bind to highly branched glycans, thereby crosslinking membrane-bound glycoproteins, which increases their lifespan [53,54]. These mechanisms have been experimentally investigated for several crucial glycosylated cell surface molecules (e.g., the epidermal growth factor receptor and E-cadherin) [54], but not in detail for ABCB1 so far. One study reported that a knockdown of *galectin-3* resensitized multidrug-resistant Caco-2 colon cancer cells to epirubicin by inhibiting ABC transporters [55]. However, the authors of this study did not attribute the effect to the property of galectin-3 to crosslink glycoproteins. Strikingly, a large-scale mass spectrometry study observed a direct interaction between galectin-3 and ABCB1 [56]. In summary, studying the relation between galectins and ABC transporters might represent a viable direction for future investigations.

In conclusion, the present results indicate that ABCB1 primarily mediates DTX resistance in DU145 and PC-3 DTXR cells and that a significant involvement of other ABC transporters is unlikely. Consequently, different approaches targeting ABCB1 to overcome DTX resistance were evaluated. Inhibition of ABCB1 with small molecule inhibitors and siRNA-mediated knockdown were most effective in resensitizing both DTXR sublines to DTX. ABCB1 exhibited additional glycosylation in DTXR cells not present in treatment-naïve control cells. A direct influence of the glycosylation status on DTX tolerance and thus, presumably on ABCB1 activity was only observed in DU145 DTXR cells, but not in PC-3 DTXR cells, indicating a cell line-dependent effect. Therefore, it would be interesting to evaluate a broader spectrum of PCa cells including AR-positive cell lines such as LNCaP regarding the involvement of ABCB1 in DTX resistance. Overall, systemic effects pose the biggest challenge in targeting overexpressed ABCB1 in PCa cells since all three examined approaches are not specific to ABCB1-expressing cancer cells. Nevertheless, targeting ABCB1 for reversing taxane resistance still holds great therapeutic potential. Given that targeted drug delivery can be realized, small molecule inhibitors of ABCB1 represent the most promising and feasible approach to be implemented in clinical routines.

## 4. Materials and Methods

### 4.1. Cell Culture

Human DU145 and PC-3 PCa cell lines and their age-matched DTX-resistant sublines DU145 DTXR and PC-3 DTXR were kindly provided by Dr. Martin Puhr and Prof. Dr. Zoran Culig [4]. The increased tolerance against DTX in the DTX-resistant sublines compared to the respective CTRL cells was previously shown via WST-1 viability assay (IC_50_ values for DTX: 5.9 nM in DU145 CTRL, 388 nM in DU145 DTXR, 8.2 nM in PC-3 CTRL, 305 nM in PC-3 DTXR) [5]. All cells were cultured in RPMI 1640 medium supplemented with 10% fetal bovine serum (FBS) (both Thermo Fisher Scientific, Dreieich, Germany) at 37 °C in a humidified atmosphere with 5% CO_2_. For culturing DTXR cells, the medium was supplemented with 10 nM DTX (Selleck Chemicals, Planegg, Germany) to maintain the selective pressure. Cells were passaged using 0.05% trypsin/EDTA (Thermo Fisher Scientific). Cell counting was done using the Muse^®^ Count & Viability kit on a Muse^®^ Cell Analyzer flow cytometer (both Luminex Corporation, Austin, TX, USA). Live-cell imaging was performed in an IncuCyte S3 Live-Cell Analysis System (Sartorius, Göttingen, Germany). Cells were tested regularly for mycoplasma using the Mycoalert^TM^ Mycoplasma Detection kit (Lonza, Basel, Switzerland). Before experimentation, all cell lines were authenticated by Short Tandem Repeat analysis as described before [5].

### 4.2. RNA Isolation, cDNA Synthesis and Quantitative PCR Analysis

Cells were harvested in phosphate-buffered saline (PBS) on ice using a cell scraper (Corning, Corning, NY, USA), centrifuged, resuspended in 375 µL TriFast (PeqLab, VWR, Darmstadt, Germany) and stored at −80 °C until further processing. RNA was isolated using the Direct-zol RNA Mini-Prep kit (Zymo Research, Freiburg, Germany) according to the manufacturer’s instructions. RNA concentrations were measured using a NanoDrop 2000c spectrophotometer (PeqLab). Next, 500 ng RNA were reverse-transcribed into cDNA using the reverse transcriptase enzyme Superscript II and other consumables purchased from Thermo Fisher Scientific according to the manufacturer’s instructions. Each qPCR reaction contained 1 µL cDNA (1:5), 0.5 µL specific TaqMan Gene Expression Assay (Thermo Fisher Scientific), 5 µL GoTaq Probe qPCR Master Mix (Promega, Walldorf, Germany) and 3.5 µL nuclease-free water. The following specific TaqMan Assays, which contained the appropriate primers and probes, were used: ABCA3 (Hs00184543_m1), ABCB1 (Hs00184500_m1), ABCB4 (Hs00983957_m1), ABCB11 (Hs00994811_m1), ABCC1 (Hs01561483_m1), ABCC2 (Hs00960489_m1), ABCC10 (Hs01056200_m1) and ABCG2 (Hs01053790_m1) as well as HPRT1 (Hs02800695_m1), RPLP0 (Hs00420895_gH) and TBP (Hs00427620_m1) as references. Then, two independent qPCR measurements per sample were performed on a LightCycler 480 Real-Time PCR System (Roche, Mannheim, Germany) with the following conditions: initial denaturation at 95 °C for 10 min followed by 45 cycles at 95 °C for 15 s and 60 °C for 60 s. CPs were determined by the automatic second derivative method and then averaged. If both CP values deviated > 0.5, the measurement was repeated. The detection limit was indicated by a CP ≥ 35. Relative expression levels were calculated via ΔCP with normalization to the geometric mean of the reference genes and n-fold expression was determined with the ΔΔCP method using the respective control.

### 4.3. Western Blot Analysis

Cells were harvested in PBS on ice by scraping, centrifuged and lysed in radioimmunoprecipitation assay (RIPA) buffer containing 1% Triton X-100, protease and phosphatase inhibitor cocktails (all Sigma-Aldrich, trademark of Merck, Darmstadt, Germany). Lysates were shock-frozen in liquid nitrogen and homogenized on a shaker at 4 °C for 30 min. The debris was removed by centrifugation and the protein concentration was determined using a Pierce™ BCA Protein Assay kit (Thermo Fisher Scientific) according to the manufacturer’s instructions. LDS sample buffer (Thermo Fisher Scientific) and 1% β-mercaptoethanol (Sigma-Aldrich) were added before incubating the samples at 60 °C for 10 min. Protein separation was performed with the NuPAGE system (Thermo Fisher Scientific) and 10–20 µg protein per lane was loaded depending on the experiment. Proteins were blotted with the iBlot semi-dry blotting system (Thermo Fisher Scientific). The membrane was blocked with Tris-buffered saline (TBS) buffer containing 0.1% tween (TBS-T) and 10% non-fat dry milk (NFDM) for 1 h at room temperature (RT). After washing with TBS-T, the membrane was probed with the respective primary antibody (Table 1) diluted in TBS-T containing 1% NFDM at 4 °C overnight. Primary antibodies were detected using horse radish peroxidase (HRP)-conjugated secondary antibodies (Anti-Rabbit-HRP: 1:2000, P0217; Anti-Mouse-HRP: 1:5000, P0260; both Dako, Hamburg, Germany) diluted in TBS-T containing 1% NFDM for 1–2 h at RT. WesternBright™ Sirius ECL (Advansta, San Jose, CA, USA) was used to detect HRP in a MicroChemi 4.2 imaging system (DNR Bio-Imaging Systems, Jerusalem, Israel). Cropping and densitometrical analysis of Western blot images was performed using ImageJ [57]. Target protein levels were normalized to the indicated reference proteins.

### 4.4. Flow Cytometry

Cells were harvested using 500 U/mL accutase in PBS (Sigma-Aldrich). Cells were centrifuged and resuspended in a staining buffer consisting of PBS with 5% FBS, 1 mM EDTA (Sigma-Aldrich), 10 mM HEPES (Sigma-Aldrich). Then, 1 × 10^5^ cells per sample were stained with anti-human ABCB1 (348606, clone UIC2, 10 µg/mL; Biolegend, San Diego, CA, USA), anti-human ABCG2 (332008, clone 5D3, 5 µg/mL; Biolegend) or corresponding isotype controls (IgG2a: 400214, clone MOPC-173, IgG2b: 400314, clone MPC-11; Biolegend) conjugated with phycoerythrin (PE). After incubation for 30 min at 4 °C in the dark, the cells were centrifuged and then resuspended in a staining buffer containing 0.5 µg/mL 7-aminoactinomycin (Biolegend) for live/dead cell discrimination. The measurement was performed on a BD Celesta flow cytometer (BD Biosciences, Heidelberg, Germany) with excitation optics containing a violet (405 nm), blue (488 nm) and yellow-green (561 nm) laser. Flow cytometry data were analyzed using FlowJo software (BD Biosciences).

### 4.5. Treatment Concentrations of ABCB1 Inhibitors, siRNA or Tunicamycin

The optimal treatment concentrations for the ABCB1 inhibitors elacridar and tari-quidar as well as for the glycosylation inhibitor tunicamycin were determined by dose-response experiments (Appendix A). The finally applied concentrations of each substance are summarized in Table 2.

### 4.6. Rhodamine 123 Efflux Assay

For evaluation of ABCB1 inhibitors, cells were seeded in black 96-well plates suitable for fluorescence measurements (six technical replicates per treatment) and cultured for 24–48 h. Then, cells were treated with RPMI 1640 medium supplemented with 0.5 µg/mL of the fluorescent ABCB1 substrate rhodamine 123 (Rh123; J&K Scientific, Beijing, China) for 15 min alone or in combination with elacridar or tariquidar. After removal of the treatment and repeated washing with PBS, RPMI 1640 medium alone or supplemented with either elacridar or tariquidar was added to the cells. Immediately thereafter, cells were imaged in intervals of 10 min over 3 h using the IncuCyte S3 Live-Cell Analysis System (10× objective, green fluorescence, 1 image/well).

In order to evaluate the influence of the glycosylation status on the ABCB1 efflux activity, cells were seeded in black 96-well plates in medium supplemented with or without tunicamycin (six technical replicates each). After 96 h, cells were treated with RPMI 1640 medium supplemented with 0.5 µg/mL Rh123 for 15 min. After removal of the treatment, repeated washing with PBS and addition of fresh RPMI 1640 medium, cells were imaged in intervals of 10 min over 1 h using the IncuCyte S3 Live-Cell Analysis System as described above.

### 4.7. Treatment with ABCB1 Inhibitors

For the WST-1 and crystal violet assays, cells were seeded in triplicates in 96-well plates and cultured to ~50% confluence. Then, cells were treated with a serial dilution of DTX (0.1 nM–10 µM) either alone or in combination with ABCB1 inhibitors. After 24 h, the treatment was substituted with fresh cell culture medium and 24 h later, the respective assay was performed.

To assess colony formation ability, cells were seeded in 6-well plates (20,000 per well for DU145 and 30,000 per well for PC-3), cultured for 96 h and then treated with 10 nM DTX and 50 nM elacridar alone or in combination for 24 h. Finally, the respective treatment was substituted with fresh cell culture medium and incubated for another 24 h followed by the conduction of the colony formation assay.

### 4.8. Treatment with ABCB1 siRNA

For the WST-1 and crystal violet assays, cells were seeded in triplicates in 96-well plates as described above. After 48 h, cells were transfected in OptiMEM containing 0.15% Lipofectamine 2000 and either siRNA against *ABCB1* (siR-ABCB1: HSS182278, sequence: 5′-GCA GCU UAU GAA AUC UUC AAG AUA A-3′) or a negative control construct (siR-CON: 12935200; all Thermo Fisher Scientific) for 4 h. Following addition of RPMI 1640 medium, the cells were cultured for 96 h and then treated with DTX (3 nM–1 µM) for 24 h. The treatment was substituted with fresh cell culture medium and 24 h later, the respective assay was performed.

For qPCR and Western blot analysis, cells were seeded in 6-well plates, cultured for 48 h and then transfected as described above. After culturing for 96 h, cells were harvested for subsequent isolation of RNA and protein as described before.

To assess the colony formation ability, cells were seeded in 6-well plates, cultured for 48 h and then transfected as described above. Thereafter, cells were cultured for 96 h and then treated with 50 nM DTX for 24 h. Finally, the respective treatment was substituted with fresh cell culture medium and incubated for another 24 h followed by the conduction of the colony formation assay.

### 4.9. Treatment with Tunicamycin

For experiments evaluating the effect of glycosylation inhibition on DTX resistance, cells were pre-treated with RPMI 1640 medium containing tunicamycin for 96 h. For Western blot analysis, the cells were seeded and pre-treated in 10 cm dishes and after 96 h of incubation, proteins were extracted as described above. For dose-response experiments via WST-1 and crystal violet assays, the cells were seeded in triplicates in 96-well plates as described above and then pre-treated with tunicamycin. After 96 h, the cells were incubated with a serial dilution of DTX (3 nM–10 µM) for 24 h. Then, the treatment was substituted with fresh cell culture medium and 24 h later, the respective assay was performed.

### 4.10. WST-1 Assay to Assess Cellular Viability

Following treatment, 10 µL of WST-1 (Roche) were added per well in 96-well plates. WST-1 is metabolized to formazan by the mitochondrial succinate-tetrazolium-reductase system. The assay measures the degree of energy generation via oxidative phosphorylation in metabolically active cells. After 60 min, formazan absorbance was measured at 450 nm with a reference measurement at 620 nm using a Berthold Mithras LB940 microplate reader (Berthold Technologies, Bad Wildheim, Germany).

### 4.11. Crystal Violet Assay to Assess Total Adherent Cell Mass

Cells in 96-well plates were fixed with methanol and stained with an aqueous solution of 0.1% crystal violet (Merck) for 10 min. After thorough washing with water, stained adherent cells were dissolved by adding 100 µL of 0.1 M sodium citrate in 50% ethanol per well and incubating for 30 min on a shaker. Crystal violet absorption was measured at 590 nm using a Berthold Mithras LB940 microplate reader.

### 4.12. Colony Formation Assay to Assess Clonogenic Properties and Long-Term Proliferation

Colony formation assays were performed to examine clonogenic properties after treatment with DTX in combination with elacridar and siR-ABCB1, respectively. Twenty-four hours after the end of treatment, the cells were harvested and colony formation assays were seeded in triplicates in 6-well plates (100 cells/well for DU145, 200 cells/well for PC-3). After 10–12 d, the colonies were fixed with methanol and then stained with an aqueous solution of 0.1% crystal violet. Colonies were imaged with the Keyence All-in-One Fluorescence Microscope BZ-X800 (4× objective, brightfield; Keyence, Neu-Isenburg, Germany) and then counted. Only colonies with > 50 cells were considered.

### 4.13. Digestion with PNGase F

Cells were harvested and protein lysates were obtained as described above. After determination of the protein concentration, 1% β-mercaptoethanol was added and samples were incubated for 10 min at 60 °C. Then, the samples were incubated with 10 mU PNGase F (Promega) per 100 µg protein on a shaker at 37 °C for 2 h with gentle shaking. Thereafter, LDS sample buffer was added and samples were stored at −20 °C until Western blot analysis.

### 4.14. Immunofluorescence Staining

For immunofluorescence staining of the ABCB1 protein, the DU145 and PC-3 DTXR cells were pre-cultured with tunicamycin for 96 h as described above, while untreated CTRL and DTXR cells served as control. Thereafter, the cells were seeded on sterile tissue culture coverslips (Sarstedt, Nümbrecht, Germany), which were placed into the wells of a 24-well plate, and cultured to ~70% confluence. Then, the cells were fixed with 4% formaldehyde, permeabilized (1% Triton X-100 in PBS), blocked with blocking buffer (1% bovine serum albumin in PBS) and finally incubated with a primary antibody against ABCB1 (1:100, MA5-13854, clone F4; Thermo Fisher Scientific) diluted in blocking buffer overnight at 4 °C. The next day, the cells were treated with the secondary antibody (Alexa Fluor 488 goat anti-Mouse IgG: 1:1000, A-11001; Thermo Fisher Scientific) diluted in blocking buffer for 1–2 h at RT. Finally, cell nuclei were stained with 1 µg/mL DAPI (AppliChem, Darmstadt, Germany) for 2–5 min before the coverslips were mounted on a microscope slide and sealed with nail polish. Immunofluorescence images were taken with the Keyence All-in-One Fluorescence Microscope BZ-X800 (40× objective, green and blue fluorescence).

### 4.15. Statistics, Data Visualization and Software

Data plotting, curve fitting, calculation of IC_50_ values and statistical tests were done in GraphPad Prism 9 (GraphPad Software, San Diego, CA, USA). When evaluating the response to DTX treatment based on WST-1 or crystal violet assay, the data were first normalized to the indicated control and then log-transformed. Non-linear ‘log(inhibitor) vs. response’ regression with variable slope was used for curve fitting and calculating the IC_50_ values. Due to data normalization, the upper plateaus of the inhibition curves were constrained to equal 1, while the lower plateaus were constrained to be ≥ 0.

All experiments conducted in this study were performed with at least three (up to six) biological replicates and data are depicted as mean ± SD if not otherwise stated. All statistical tests were done without the assumption of equal variance. Non-paired t-tests with Welch’s correction were performed to test for the statistical significance of differences between two groups. A one-sample t-test was used to compare normalized data with their respective controls. *p* values < 0.05 were considered statistically significant.

## Figures and Tables

**Figure 1 ijms-24-00666-f001:**
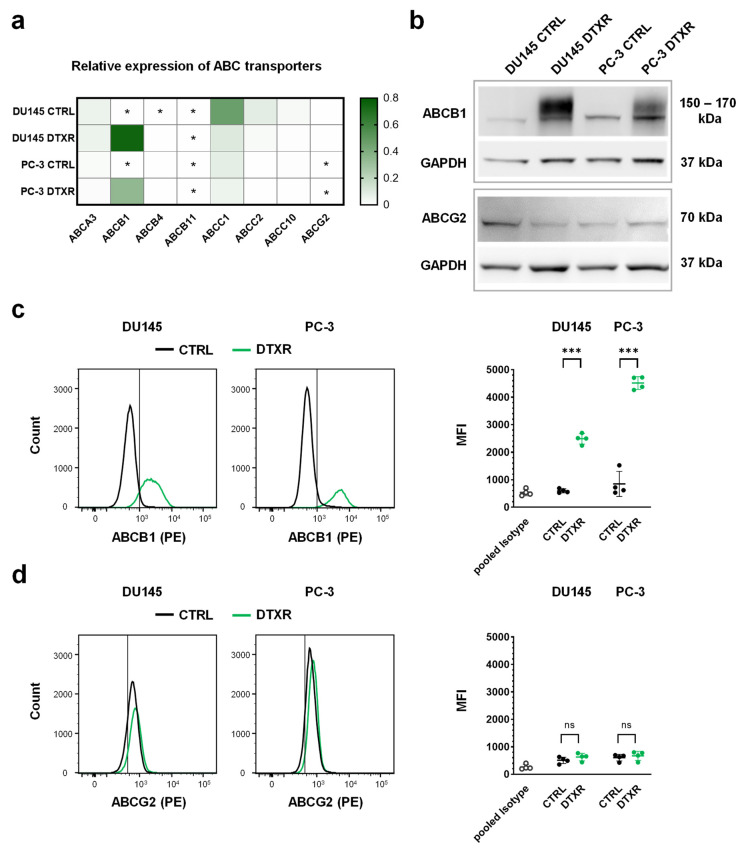
RNA and protein expression of selected ABC transporters, which are known to be involved in taxane resistance and/or MDR, in DU145 and PC-3 DTXR cells compared to CTRL cells. (**a**) The relative transcript levels of eight selected ABC transporters were plotted on a heat map. Cells with an expression of the respective ABC transporter beyond the detection threshold (CP ≥ 35) are marked with an asterisk (*). (**b**) Exemplary Western blots for detecting ABCB1 and ABCG2 proteins in CTRL and DTXR cells. GAPDH served as a reference protein. (**c**,**d**) Exemplary flow cytometry analysis for (**c**) ABCB1 and (**d**) ABCG2 proteins via specific antibodies conjugated with the fluorescent phycoerythrin (PE) in CTRL and DTXR cells. The vertical black lines indicate the threshold of unspecificity according to the pooled isotype controls. The adjacent graphs to the right depict the median fluorescence intensities (MFI) for ABCB1 and ABCG2 protein, respectively. Data are depicted as mean ± standard deviation (SD) of three to four independent biological replicates. Non-paired t-test with Welch’s correction was performed for comparison of CTRL vs. DTXR cells: *** *p* < 0.001, ns: not significant.

**Figure 2 ijms-24-00666-f002:**
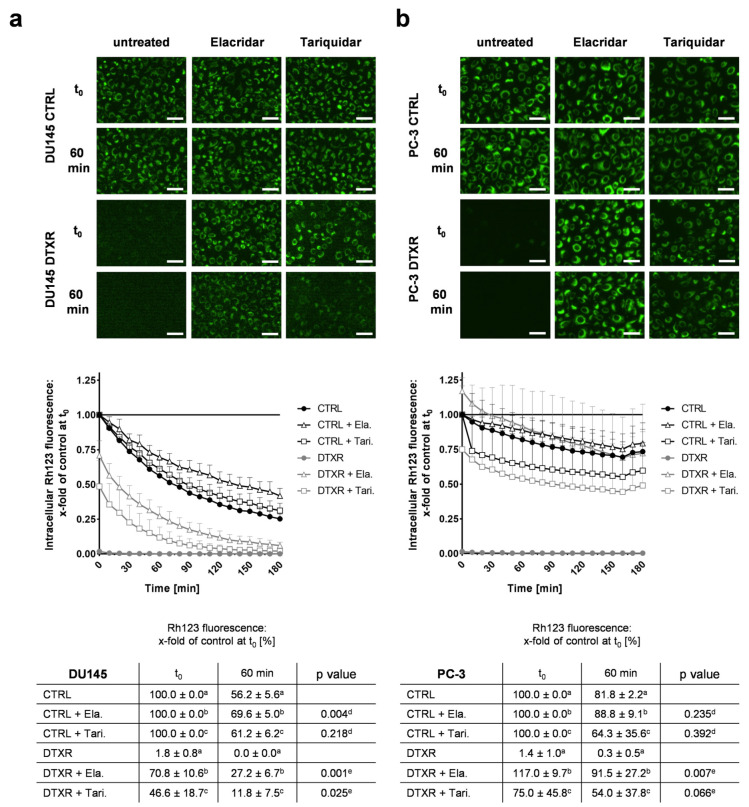
Influence of ABCB1 inhibitors elacridar and tariquidar on ABCB1 efflux activity of the fluorescent ABCB1 substrate Rh123 in DU145 and PC-3 DTXR cells compared to CTRL cells. (**a**) DU145 and (**b**) PC-3 cells were treated with 0.5 µg/mL Rh123 in the presence or absence of an ABCB1 inhibitor (50 nM elacridar or tariquidar) for 15 min. After several washing steps, RPMI 1640 medium with or without one of the inhibitors was added and subsequently the cells were imaged in 10 min intervals. As examples, the first image (t_0_) and an image taken after 60 min are depicted for CTRL and DTXR cells (scale bar 50 µm). Longitudinal data were normalized to the corresponding treatment of CTRL cells at t_0_ (indicated by ^a^, ^b^ or ^c^ in the table; black line in the graph). Data are depicted as the mean of four to five independent biological replicates + SD in the graphs for better readability or ± SD in the tables. Non-paired t-test with Welch’s correction was performed for comparison of ^d^ CTRL + elacridar/tariquidar vs. CTRL cells and ^e^ DTXR + elacridar/tariquidar vs. DTXR cells at 60 min. Ela.: elacridar, Tari.: tariquidar.

**Figure 3 ijms-24-00666-f003:**
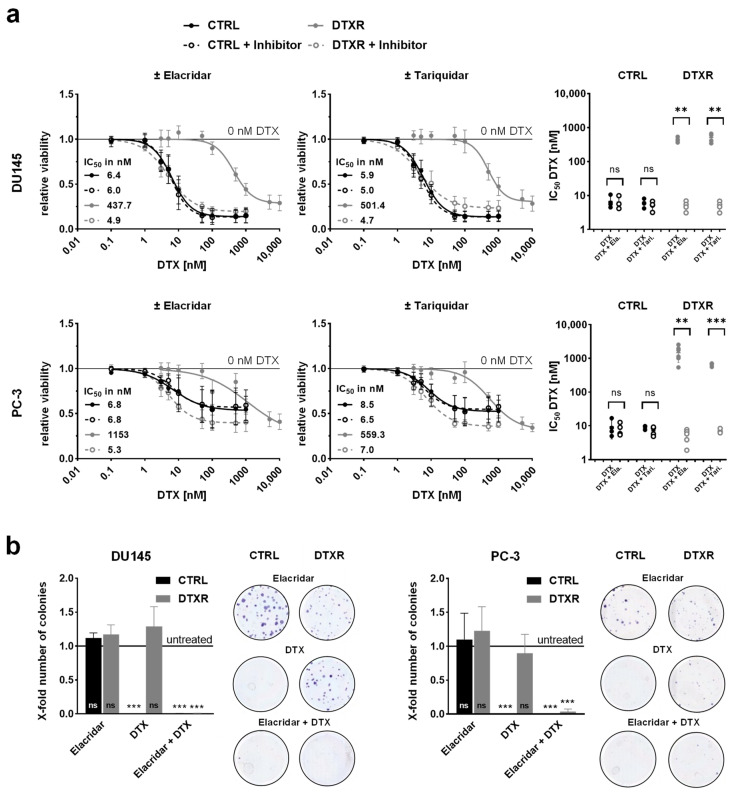
Influence of the ABCB1 inhibitors elacridar and tariquidar in combination with DTX on cellular viability and clonogenic survival of DU145 and PC-3 DTXR cells compared to CTRL cells. (**a**) DU145 and PC-3 CTRL and DTXR cells were treated with a serial dilution of DTX in the presence or absence of an ABCB1 inhibitor (50 nM elacridar or tariquidar). After treatment, the metabolic activity was measured by the WST-1 assay and IC_50_ values for DTX were calculated. The adjacent graphs to the right depict IC_50_ values calculated from individual experiments. Ela.: elacridar, Tari.: tariquidar. (**b**) CTRL and DTXR cells were treated with either 50 nM elacridar or 10 nM DTX alone or with a combination of both. Thereafter, cell colony formation assays were seeded and cell colonies were counted after 10–12 d. Exemplary colonies are shown in the right panels. All data were normalized to the indicated control (black line). Data are depicted as mean ± SD of four to six independent biological replicates. (**a**) Non-paired t-test with Welch’s correction for comparison of two treatment groups and (**b**) one-sample t-test for comparison to the untreated control: ** *p* < 0.01, *** *p* < 0.001, ns: not significant.

**Figure 4 ijms-24-00666-f004:**
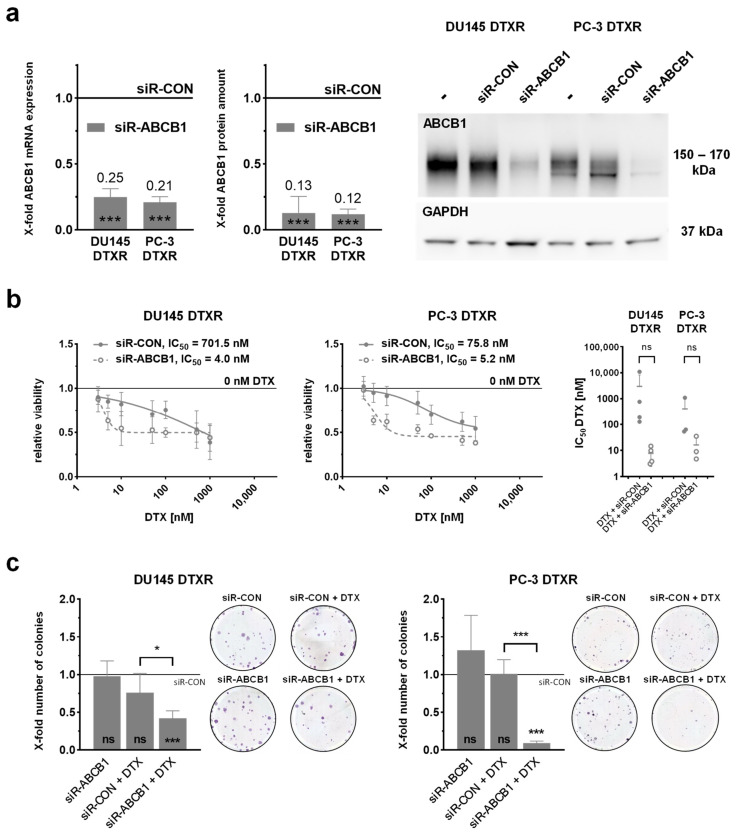
Influence of siRNA-mediated knockdown of *ABCB1* in combination with DTX on cellular viability and clonogenic survival of DU145 and PC-3 DTXR cells. (**a**) The relative transcript and protein levels of ABCB1 in DTXR cells 96 h after siRNA transfection start are depicted. An exemplary Western blot for the detection of ABCB1 protein with GAPDH as reference protein is shown. (**b**) DU145 and PC-3 DTXR cells were treated with a serial dilution of DTX after transfection with either siR-CON or siR-ABCB1 (20 nM). After treatment, the metabolic activity was measured by the WST-1 assay and IC_50_ values for DTX were calculated. The adjacent graphs to the right depict IC_50_ values calculated from individual experiments. (**c**) DTXR cells were transfected with either siR-CON or siR-ABCB1 (20 nM) and 96 h after transfection start the cells were treated with 50 nM DTX. Thereafter, cell colony formation assays were seeded and cell colonies were counted after 10–12 d. Exemplary colonies are shown in the right panels. All data were normalized to the indicated control (black line). Data are depicted as mean ± SD of three to five independent biological replicates. (**a**,**c**) One-sample t-test for comparison to the siR-CON-treated controls and (**b**,**c**) non-paired t-test with Welch’s correction for comparison of two treatment groups: * *p* < 0.05, *** *p* < 0.001, ns: not significant.

**Figure 5 ijms-24-00666-f005:**
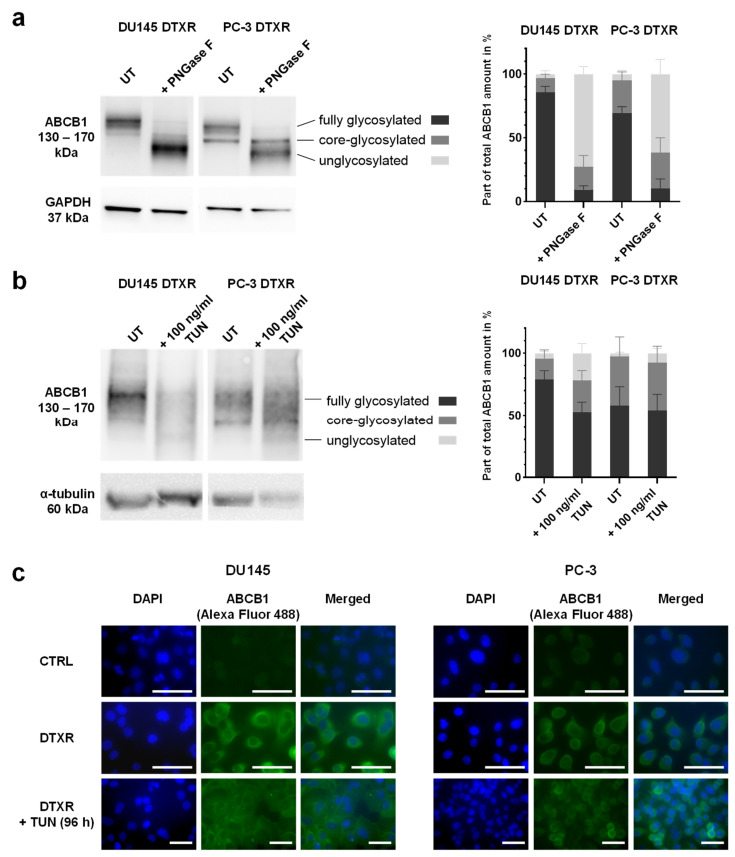
Influence of PNGase F digestion and tunicamycin treatment on the glycosylation status and localization of ABCB1 protein in DU145 and PC-3 DTXR cells. (**a**) Protein lysates of DU145 and PC-3 DTXR cells were treated with PNGase F for 2 h and compared to untreated protein lysates (UT). An exemplary Western blot for the detection of ABCB1 protein with GAPDH as reference protein is shown. (**b**) DU145 and PC-3 DTXR cells were pre-treated with 100 ng/mL tunicamycin (TUN) for 96 h and compared to untreated cells (UT). An exemplary Western blot for detecting ABCB1 protein with α-tubulin as reference protein is shown. (**a**,**b**) The protein bands were analyzed densitometrically and then normalized to the respective reference protein. The different glycosylation statuses are depicted as part of the total ABCB1 amount in %. (**c**) Immunofluorescence staining of ABCB1 in DU145 and PC-3 CTRL cells as well as DTXR cells with or without pre-treatment with 100 ng/mL tunicamycin (TUN) for 96 h. Cell nuclei were visualized with DAPI (blue), ABCB1 was detected using an anti-ABCB1 antibody and an Alexa Fluor 488-conjugated secondary antibody (green) and then the images were merged (scale bar 40 µm). Exemplary images and data, which are depicted as mean ± SD, are taken from four to five independent biological replicates.

**Figure 6 ijms-24-00666-f006:**
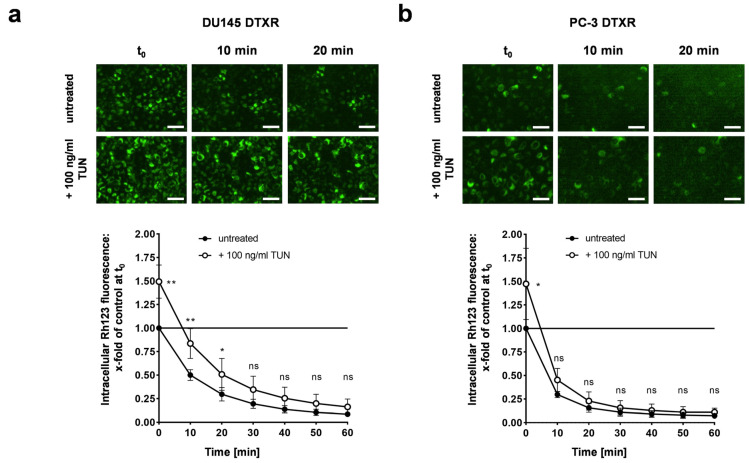
Influence of the glycosylation status of ABCB1 protein on ABCB1 efflux activity of the fluorescent ABCB1 substrate Rh123 in DU145 and PC-3 DTXR cells. (**a**) DU145 and (**b**) PC-3 DTXR cells with or without pre-treatment with 100 ng/mL tunicamycin (TUN) for 96 h were incubated with 0.5 µg/mL Rh123 for 15 min. After several washing steps, RPMI 1640 medium was added and subsequently the cells were imaged in 10 min intervals. As examples, the first image (t_0_) and images taken after 10 and 20 min are depicted for DTXR cells with or without tunicamycin pre-treatment (scale bar 50 µm). Longitudinal data were normalized to DTXR cells without tunicamycin pre-treatment at t_0_ (black line in the graph). Data are depicted as mean ± SD of four to five independent biological replicates. Non-paired t-test with Welch’s correction was performed for comparison of DTXR cells with tunicamycin pre-treatment vs. DTXR cells without tunicamycin pre-treatment at each timepoint: * *p* < 0.05, ** *p* < 0.01, ns: not significant.

**Figure 7 ijms-24-00666-f007:**
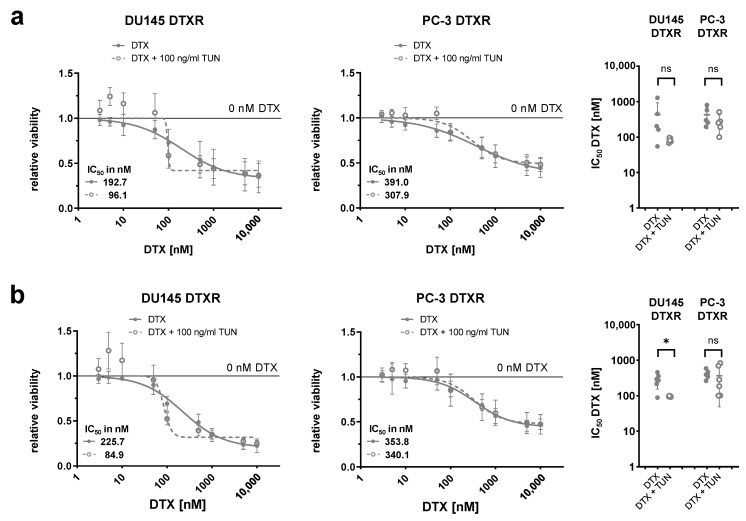
Influence of tunicamycin-mediated deglycosylation of ABCB1 protein in combination with DTX on cellular viability of DU145 and PC-3 DTXR cells. DU145 and PC-3 DTXR cells with or without pre-treatment with 100 ng/mL tunicamycin (TUN) for 96 h were treated with a serial dilution of DTX. After treatment, the metabolic activity and the total adherent cell mass were determined by (**a**) the WST-1 assay and (**b**) the crystal violet assay, respectively, and IC_50_ values for DTX were calculated. The adjacent graphs to the right depict IC_50_ values calculated from individual experiments. All data were normalized to the indicated control (black line). Data are depicted as mean ± SD of four to six independent biological replicates. Non-paired t-test with Welch’s correction for comparison of two treatment groups: * *p* < 0.05, ns: not significant.

**Table 1 ijms-24-00666-t001:** Primary antibodies for Western blot analysis.

Primary Antibody against	Source	Catalog No., Clone	Dilution
ABCB1	Cell Signaling Technology (Leiden, The Netherlands)	12683, D3H1Q	1:1000
ABCG2	Cell Signaling Technology	42078, D5V2K	1:1000
GAPDH	OriGene Technologies (Rockville, MD, USA)	5G4-6C5, 6C5	1:5000
α-tubulin	Cell Signaling Technology	3873, DM1A	1:3000

**Table 2 ijms-24-00666-t002:** Applied doses and targets of inhibition of small molecule inhibitors and siRNA used in this study.

Reagent	Inhibition of	Applied Dose
Elacridar	ABCB1 & ABCG2 protein	50 nM
Tariquidar	ABCB1 & ABCG2 protein	50 nM
siRNA (siR-ABCB1)	ABCB1 mRNA	20 nM
Tunicamycin	Glycosylation in the ER	100 ng/mL

## Data Availability

The data presented in this study are available on request from the corresponding author.

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
