# Peer review of "Comprehensive Evaluation of Multiple Approaches Targeting ABCB1 to Resensitize Docetaxel-Resistant Prostate Cancer Cell Lines"

_ijms, 2022, doi:10.3390/ijms24010666_

Round 1

Reviewer 1 Report

The manuscript assessed multiple approaches in targeting ABCB1 to resensitize docetaxel-resistant PCa cells to docetaxel. The manuscript is well written, substantiates earlier findings, and presents some novel findings. While several studies have shown the importance of ABCB1 in docetaxel resistance previously, this manuscript adds to the knowledge by examining the effect of glycosylation on the activity of ABCB1. A few comments are appended below. 

Major comments:

1) The authors used AR-ve cell lines DU145 and PC-3 for this study. Clinically, the percentage of PCa cells that are completely -ve for AR has been shown to be small. The results would be more meaningful if the study included some AR+ve DTXR cell lines. 

2)  Given that nearly half of the results have been reported previously, the manuscript would benefit form further examination of the effect of glycosylation on ABCB1. Please test whether treatment with tunicamycin leads to ubiquitination and subsequent degradation of ABCB1. Similarly, please examine whether reduction in glycosylation reduces the activity of ABCB1 in drug efflux (rhodamine). These experiments will improve our understanding of the role of glycosylation of ABCB1. 

3) Given that tunicamycin is not specific to ABCB1, how would you confirm that the observed effects in Fig. 5C and 6 are due to the deglycosylation of ABCB1?

4) Previous studies have shown that p53-status may be important in the response to docetaxel. Please discuss/examine whether the differential effect of tunicamycin on DU145-DTXR and PC-3-DTXR is due to their respective p53-status (p53-mutant vs. p53-null). 

Minor comments: 

1) In all figures, please state the number of biological and technical replicates in each experiment. 

2) Fig. 2 is missing significance level and error bars. 

Reviewer 2 Report

The present article submitted by Linke et al., titled “Comprehensive Evaluation of Multiple Approaches Targeting ABCB1 to Resensitize Docetaxel-Resistant Prostate Cancer Cell Lines” explored the Multiple Approaches such as using inhibitor, siRNA mediated and post translational glycosylation to targeting ABCB1 to resensitize docetaxel-resistant prostate cancer cell. The manuscript is well designed, experiment performed and drafted properly. The abstract is well written and conclusive. The figures are very clear and informative. There are some changes need to be done in the reference section. Some of the references are not in a format. For ex.; Ref 6, 38, 38, 41, 52 etc. The title is in the “capitalized each word” form. It should be “sentence case”. Authors should provide doi of each references. Some of the reference doi’s are missing.

Reviewer 3 Report

In the manuscript "Comprehensive evaluation of multiple approaches targeting ABCB1 to resensitize docetaxel-resistant prostate cancer cell lines" the authors applied different strategies to assess the impact of ABCB1 in docetaxel resistance.

Overall the manuscript is well organized and structured, the figures are readable, and the discussion integrates relevant info on taxane resistance. However, some aspects need to be revised and improved over the manuscript:

1. Over the manuscript, the IC50 to DTX of resistance cell lines is not mentioned and this should be included to understand the reversion studies better. Despite the author's including the reference to DTX resistance cell lines developing paper (ref 5), this IC50 info should be included, for example, in the Material section.

2. The author explored the gene expression levels of 8 ABC transporters and then decide to study only ABCB1 and ABCG2 at protein levels by WB and flow cytometry. The selection of ABCG2 study need to be explained since no differences in gene expression were observed between senstive and resistant models. Moreover, ACBC family presented differences between cell lines that should be confirmed in terms of protein expression as performed for ABCG2.

3. In figure 1, to impove figure readability the other could use different colors for the conditions rather than grey scale. 

4. In section 2.1 of the results the authors mention the differences in terms of % of ABCB1-positive cells but do not make any mention to the graphs c & d of figure 1 that presented significant differences in MFI between cell lines. This should be revised in manuscript. 

5. In line with previous mention in poin 1, for the intrepertation of siRNAs studies is importnat to have present the IC50 of DTX in each cell line. After consulting the reference 5, IC50 DTX is 388nM for DU145 DXTR and 305 nM to PC3 DTXR. Analysing the siRNA results, in DU145 cell line the control construction (siR-CON) have an antagonic effect with DTX since the IC50 increase to 701nM. However, in PC-3 model the siR-CON leads to a significant reduction on DTX IC50 (reduction 4x the IC50). The author should investigate the mechanism that could justify this effect on DTX sensitivity.

6. The author hypothesizes that the fluorescent foci observed in DU145 DTXR cells may be ABCB1 accumulation in some cell compartments, as ER. The authors should included an ER marker in immunofluorecence studies and evaluate the ABCB1 accumulation. 

7. The authors did not assess if the glycosylate state affects the activty of ABCB1. This aspect should be assess since no reversion of DTX IC50 was saw when combined with TUN in PC3.

Round 2

Reviewer 3 Report

The authors answer the raised points and improved the quality of images. I have no further comments.